# Intraocular Pressure Fluctuation in Primary Open-Angle Glaucoma with Canaloplasty and Microcatheter Assisted Trabeculotomy

**DOI:** 10.3390/jcm11247279

**Published:** 2022-12-08

**Authors:** Chen Xin, Ningli Wang, Huaizhou Wang

**Affiliations:** Glaucoma Department, Beijing Tongren Eye Center, Beijing Tongren Hospital, Capital Medical University, Beijing 100730, China

**Keywords:** primary open angle glaucoma, Schlemm’s canal, IOP fluctuation, canaloplasty, trabeculotomy

## Abstract

Background: Schlemm’s canal (SC) targeted procedures constitute a promising therapy for open angle glaucoma (POAG), safer and less invasive. However, little attention was paid to the intraocular pressure (IOP) variation in patients receiving these procedures, which is the risk factor for POAG progression. This study is to evaluate the IOP variation in eyes with POAG after modified canaloplasty (MC) and microcatheter assisted trabeculotomy (MAT). (2) Methods: POAG with good IOP in office hours after MC or MAT and age-matched normal subjects were recruited in this prospective coherent study. IOP in sitting and supine positions and 24-h IOP was measured. Aqueous vein and blood reflux into the SC were examined. (3) Results: Among 20 normal subjects, 25 eyes with MC eyes and 30 eyes with MAT were recruited in this study. Aqueous veins are frequently located in the inferior nasal quadrants in all groups. No pulsatile signs were observed in an aqueous vein in the MAT group but they were observed in 68% of the MC group. Blood reflux in the SC could be seen in all the operated eyes. The IOP in the sitting position was not significant different among groups (*p* = 0.419). Compared to normal, the IOP increased dramatically after lying down for 5 min in the MC and MAT groups (PMC vs. normal = 0.003, PMAT vs. normal = 0.004), which is similar for IOP change after lying down for 60 min (PMC vs. normal < 0.001, PMAT vs. normal < 0.001). In terms of diurnal IOP, subjects were stable in the MAT group (*p* < 0.01), variable in the normal group (*p* = 0.002), and most fluctuant in MC group (*p* < 0.001). (4) Conclusions: MC and MAT reduce the IOP but present aberrant short-term IOP regulation, which should be paid attention to in clinical settings.

## 1. Introduction

Glaucoma is a leading cause of irreversible blindness worldwide [1]. Lowering intraocular pressure (IOP) is the only practical way to control the progression of the disease [2]. Trabeculectomy is still considered as the most efficient surgical treatment for glaucoma, reducing IOP to low teens and damping IOP fluctuations induced by postural changes and diurnal rhythms [3,4]. However, due to vision-threatened complications as well as long-standing bleb related issues [5,6], trabeculectomy is a second choice to medication and is mainly applied to advanced or late-stage glaucoma cases.

There are now surgical options aiming to recover the natural aqueous pathway for primary open angle glaucoma (POAG) [7,8]. Canaloplasty [9,10] and microcatheter assisted trabeculotomy (MAT) [11] are two representatives of Schlemm’s canal (SC) based glaucoma surgery. Many studies have shown that SC-based surgeries result in a modest IOP reduction and exhibit less complications than trabeculectomy [12,13], and they have been suggested as efficient choices for both mild and moderate POAG cases.

The integrity and synergic motion of tissue in the trabecular meshwork (TM) pathway plays an important role in regulating the drainage of aqueous humor and normality of IOP fluctuation, such as the natural resistance built-up by the juxtacanalicular tissue and SC endothelium, the TM-SC configuration relationship [14,15] and the segmental space of the SC lumen [16]. For one thing, canaloplasty and MAT reduce the resistance in SC inner wall and juxtacanalicular region, resulting in IOP reduction. For another, both procedures break tissue linkage and disturb synergic motion, which might affect the IOP variation. IOP fluctuation is a risk factor for POAG progression and ingenious regulation of the TM pathway is a prerequisite for IOP stability. Different tissue interventions of canaloplasty and MAT might induce diverse IOP fluctuations. Some studies provided vital information on the 24-h IOP fluctuation relevant to the canaloplasty [17,18], but there have been few reports on the postural IOP changes. 

The aqueous vein, a visible indicator on the ocular surface, characterizes the regulation of aqueous outflow. It presents different patterns in glaucoma and associates with its progression [19,20]. Although canaloplasty or MAT are intended to reconstruct the aqueous outflow, their effect on the aqueous vein has never been studied. We thus conducted this study to evaluate the effects of canaloplasty and MAT on IOP fluctuations and the aqueous vein, in order to provide a greater understanding of these surgical procedures. 

## 2. Materials and Methods

### 2.1. Subjects

This cross-sectional study was approved by the Institutional Review Board of Beijing Tongren Hospital (TRECKY2018-066) and was in accordance with the tenets of the Declaration of Helsinki. Case series of POAG with successful surgical interventions (canaloplasty or MAT) and age-matched normal subjects were recruited from August 2018 to October 2019 at Beijing Tongren Eye Center. Informed consent was provided for each patient. Normal subjects were diagnosed as age-related cataract or ametropia or early-stage age related macular degeneration without any other ocular disease, traumatic and surgical history. The subjects in this study did not have any systemic diseases. The diagnosis of POAG was confirmed from clinical records as meeting the following criteria: (1) uncontrolled IOP while under maximal tolerated medical therapy (IOP > 21 mmHg or visual field deterioration with current IOP), (2) open angle by gonioscopy with no signs of congenital or acquired abnormalities, (3) typical changes in C/D and visual field test. All POAG eyes received one surgical treatment, either canaloplasty or MAT, without any recorded complications at least 1 year prior to the initiation of the study. All the recorded postoperative IOP values (1-week, 2-week, 1-month, 3-month, 6-month and 12-month) were no more than 18 mmHg without medication. POAG eyes with canaloplasty had no detectable bleb based on the image of anterior segment optical coherent tomography (AS-OCT) (Casia I, Tomy, Japan). All surgical procedures were performed by Dr. W.N.L and Dr. W.H.Z. Canaloplasty [10] and MAT [21] were performed as previously described in detail (Appendix A). All normal and POAG subjects received comprehensive ocular examinations in two clinic visits. During the first visit, IOP was measured by Goldmann applanation tonometer (GAT), central corneal thickness (CCT) was measured by AS-OCT, aqueous vein was examined by slitlamp, blood reflux into the SC was captured by gonioscope and postural IOP changes were monitored. Approximately 2 weeks after the first clinical visit, 24-h IOP monitoring was performed overnight for all recruited subjects. 

### 2.2. Aqueous Vein Examination

For a general examination using a slit lamp biomicroscope, subjects were instructed to place their jaws on the bracket and foreheads against the bar. Diffuse light was projected to the conjunctiva, providing for high magnification (40×) videos to be recorded (Appendix A). Aqueous veins presented as colorless loops in the limbus. If pulsatile stratification of the aqueous humor and blood was observed in recipient vessels, it was defined as an aqueous vein. If not identifiable, gentle digital pressure was used through the lower lid and a transient bolus of aqueous humor was triggered to move into the recipient vessel of the aqueous vein. According to the video, the number, location, and status (pulsatile or static) of identifiable aqueous veins were analyzed by Dr X.C. One month later, all the videos were reanalyzed by Dr. X.C to evaluate the intra-observer reliability. 

### 2.3. Blood Reflux into the SC

Gonioscopy was performed by Dr. X.C using a Goldmann two-mirror contact lens on all POAG cases with canaloplasty and MAT. During gonioscopy, slight pressure was applied to the viewed quadrant to drive blood reflux into the SC. The presence or absence of blood reflux in the SC was recorded.

### 2.4. Postural IOP Change

All examinations were performed by Dr. X.C during the hours of 12:00 P.M. to 2:00 P.M. The IOP was first measured in mmHg using a Tono-pen (Mentor, Norwell, MA, USA) when the patient sitting upright. Subjects were then asked to lie down on a bed, with a soft pillow underneath their head and asked to remain in a supine position to keep their eyes parallel to the bed. IOP was measured 5 min and 60 min after positional changes. Subjects were instructed to stay awake during they were in the supine position. 

### 2.5. 24-h IOP Monitoring

Monitoring was performed during overnight hospital admission. Seven IOPs were measured in mmHg in the seated position at various times (2:00 a.m., 6:00 a.m., 8:00 a.m., 10:00 a.m., 2:00 p.m., 6:00 p.m., 10:00 p.m.) using GAT for individual subjects.

### 2.6. Statistical Analysis

SPSS18.0 software (IBM, New York, NY, USA) was used for statistical analyses. Data were presented as mean ± standard deviation (SD). Paired or unpaired t-tests or repeated measures analysis of variance (ANOVA), followed by multiple comparison were analyzed by Tukey-Kramer procedures. Each comparison was labeled as significant when *p* < 0.05. 

## 3. Results

### 3.1. Demographic and Baseline Characteristics of the Subjects 

A total of 20 normal and 55 POAG subjects (25 eyes with MC and 30 eyes with MAT) were recruited in this study. No significant differences were observed amongst groups with respect to age, gender ratio, side of the operated eye, intervals between surgery and the study, maximum recorded preoperative IOP level (with medication) and the IOP measurement when subjects were recruited into the study (Table 1).

### 3.2. Aqueous Vein Examinations

The intraclass correlation coefficient of intra-observer reliability on the number and the status of the aqueous veins was 0.973 and 0.867, respectively. A total of 70 pulsatile aqueous veins, 2–5 individually, were identified in normal subjects, most of which were observed in the inferior nasal quadrants (39/70). In eyes with POAG which experienced MC, a total of 45 aqueous veins, 1–3 for each eye, were observed, and the majority of these were located in the inferior nasal quadrants (28/45). Furthermore, 32 veins in 17 eyes (68%) were static even with pressure, while others presented pulsatile flow (Figure 1). The IOP did not differ significantly between eyes with the pulsatile flow or static aqueous vein (14.8 ± 2.4 mmHg vs. 15.0 ± 2.5 mmHg, *p* = 0.731). In eyes with POAG that experienced MAT, 54 static aqueous veins were observed, 1–3 for each eye, mostly located in the inferior nasal quadrants (34/54). The aqueous veins were frequently observed in normal subjects (*p* < 0.01) but were spread between the normal and POAG subjects (*p* = 0.656). 

### 3.3. Blood Reflux into the SC

All quadrants in POAG patients that experienced canaloplasty or MAT presented blood reflux into the SC. In three eyes that experienced MAT, blood simultaneously dropped from a point in the superior region of the TM when visible blood refluxed into the SC.

### 3.4. Postural IOP Changes

The sitting IOP was similar among groups (*p* = 0.382). After 5 min, the supine IOP increased significantly by 1.9 ± 0.7 mmHg, 4.7 ± 2.4 mmHg, and 4.6 ± 1.9 mmHg in normal, MC, and MAT groups, being much higher in the MC and MAT groups (*p* < 0.001). After 60 min, the supine IOP increased significantly by 1.2 ± 1.2 mmHg, 3.4 ± 1.9 mmHg, and 3.3 ± 1.8 mmHg in the normal, MC and MAT groups; it was again much higher in the MC and MAT groups compared to the normal group. However, compared to lying down for 5 min, the supine IOP at 60 min decreased significantly in the MC and MAT groups (Figure 2).

In the MC group, after lying down, subjects with static aqueous veins presented higher IOP change (5.4 ± 1.0 vs. 4.1 ± 1.5, *p* = 0.01), whereas after lying down for 60 min, the IOP changes became similar (*p* = 0.407).

### 3.5. 24-h IOP Monitoring

The IOP fluctuations in all groups are listed in Table 2. Except for the MAT group (*p* = 0.230), the normal and MC groups presented significant fluctuations (*p* = 0.002 and <0.0001, respectively). The maximum of IOP and IOP changes were significantly different among all groups (*p* = 0.002). Compared to normal subjects, the maximum IOP was lower in the MAT group (*p* = 0.046) and the IOP changes were higher in the MC group (*p* = 0.016). In addition, the IOP peaked at 2:00 a.m. in the MC group (Figure 3).

## 4. Discussion

In this study, we selected POAG subjects with MC without bleb. Thus, both MC and MAT were treated as bleb-independent, efficient, and safe candidates for POAG cases. Multiple mechanisms have been identified underlying IOP regulation through MC, including unplugging herniations of the TM into collector channels, the formation of microperforations in the TM and SC inner wall, and restoring the potency of the SC and downstream collector channels. MAT mainly overcomes the permeability resistance in the TM and SC region to control the IOP [14,15]. The changes in the TM-SC region induced by MC and MAT may have an impact on IOP regulation. 

Aqueous veins provide an intuitive way to witness the circulation of aqueous humor. In this study, we observed the transparent aqueous content passing through the aqueous veins in a pulsatile fashion in normal subjects as reported previously [11,16]. Moreover, fewer but an optimal number of aqueous veins was identified in the eyes of POAG cases after successful MC or MAT, indicating that the observable aqueous content passing through the aqueous vein is a sign for a functional distal pathway and indicates good IOP control. However, no pulsatile aqueous flow was seen through the aqueous vein in POAG cases subjected to MAT. With the exception of 13 aqueous veins (8 eyes), no pulsatile aqueous movement was observed in the aqueous veins of POAG cases subjected to MC. Pulsation characterizes the TM pumping system [11]. The potential reasons for the disappearance of the pulsatile pattern in the aqueous veins include ablation of TM during MAT, the breakdown of intraluminal structures in the SC when the probe was passed, the biomechanical influence of the injection of the viscoelastic [17], and the circumferential inward stretching of the TM induced by the tensional suture or the elastics [18]. Considering the same surgical procedures, the characterization of aqueous veins in the eyes with canaloplasty might indicate that tensional pressure loaded by the suture interferes with the regulation of aqueous drainage. 

Blood reflux into the SC in all POAG cases that experienced successful MAT or MC is another sign of proper drainage of the distal pathway and implication of adequate IOP control. Currently, only a few studies have carried out pathological investigations on trabeculotomy. The blood dropped from a point in the original TM region in cases with MAT indicating that tissue remodeling is effectuated in the margin where TM was dissected [19].

Several studies have highlighted the importance of abnormal IOP fluctuations on the progression of POAG during nighttime and in the supine position [20]. Thus, in this study, we investigated the efficiency of successful MC and MAT procedures on postural IOP changes and diurnal IOP fluctuations. Next, we selected a rebound tonometer that could be used in different positions, and found that the values were highly reproducible. This study showed that the IOP measured by rebound tonometer was comparable to the value obtained by GAT. 

The current study confirmed that the supine posture induced IOP elevation in all subjects, and it is higher in the MC and MAT groups. The magnitude of IOP increases in POAG cases that have undergone MC is 4.3 ± 1.9 mmHg, which is similar to that reported previously [21]. It is assumed that the posture-related increase of episcleral venous pressure (EVP) and choroidal blood volume induces the IOP [22]. Moreover, postural changes are accompanied by alterations in ocular arterial blood pressure, pulsatile and non-pulsatile ocular blood flow [23], and intracranial pressure [24]. In healthy eyes, there was a 1 mmHg elevation in IOP for each 0.83 mmHg increase in EVP [22]. The increased resistance in the TM region in POAG cases may lead to higher IOP reactions to elevated EVP. Since surgical procedures restore the aqueous outflow through a natural pathway, MC and MAT significantly increase the IOP reaction to acute posture alteration and cause progressive IOP adaptation for the supine position.

In MAT, circumferentially ablated TM leads to the direct communication between the intrascleral vessels and the anterior chamber. These distal vessels in the aqueous outflow pathway assemble venule-containing receptors sensitive to pressure [25,26]. The aberrant pressure gradient alters the composition and stiffness of the wall of these vessels, which might influence the IOP reaction in response to postural changes. In MC, although the TM remains intact, the inward persistent tension on the inner wall of the SC interferes with the TM motion reactive to the pulsatile and non-pulsatile ocular blood flow, whereas in the normal condition, the SC opens segmentally and is presented as a patent lumen; its configuration changes in reaction to the IOP variation. However, it is circumferentially and lastingly opened by canaloplasty. Together, these factors may result in a large magnitude of IOP change reactive to posture alterations. Thus, it is hard to explain why POAG cases with canaloplasty and pulsatile aqueous veins presented dramatic IOP variance, which needs to be further studied.

Diurnal IOP fluctuation is another factor closely related to glaucoma progression [2] and a sign of physiological regulation of the aqueous outflow system [27]. Reportedly, diurnal IOP differs among normal subjects from 2–6 mmHg [28], which is significantly increased in POAG patients [29,30]. Herein, we showed that diurnal IOP fluctuations were reduced by MAT and magnified by MC. Previous studies reported that IOP fluctuation is reduced after trabeculotomy with trabeculotome [31] and trabeculotomy with sinusotomy [32]. The larger regions of TM ablation could be ascribed to a further reduction in diurnal IOP fluctuation compared to a trabeculotomy with trabeculotome. However, although MC achieved an IOP level to mid-teens mmHg in this study, IOP spikes were detcted at night. Partially overcoming permeability resistance in the TM and a non-physiological persistent opening of the SC lumen may influence the regulation of IOP. 

The present study has some limitations. First, the IOPs are not self-compared before and after surgical procedures. Since POAG patients always present high IOP using 3–4 different topical medications in our clinic, it is difficult to wash out the effect of the medication, which exerts a significant effect on IOP levels and IOP fluctuations. Second, only one diurnal IOP curve was available. In the current study, all diurnal IOPs were measured by a single investigator and in the same body position, which could prevent variations in the measurement. Third, the limited number of subjects in this study was taken into consideration while drawing conclusions. The number of subjects should be increased in future studies.

## 5. Conclusions

In conclusion, the current study provides novel information regarding IOP control after MC and MAT. Strikingly, MAT dampens diurnal IOP fluctuations. However, both MAT and MC induced dramatic postural IOP changes. Moreover, this study showed that aqueous veins could be identified in eyes with successful MC and MAT. This might serve as an indicator for a clinician to foresee the long-term surgical outcome and the pattern of aqueous veins may improve our understanding of these surgical procedures. Lastly, the various changes in IOP fluctuation and patterns of aqueous veins indicated the potential mechanisms of different surgical procedures. In this study, MAT stabilizes IOP. 

## Figures and Tables

**Figure 1 jcm-11-07279-f001:**
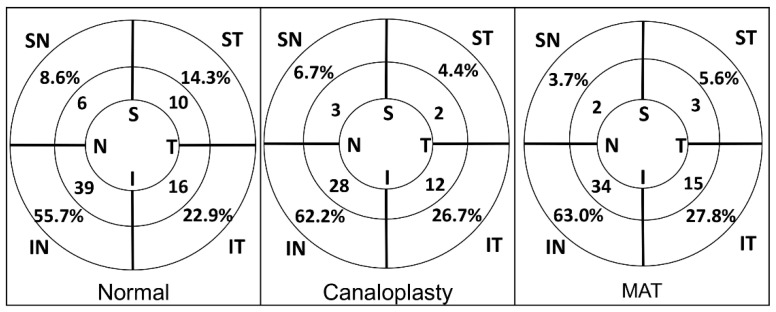
Aqueous vein distribution in eyes. Note: MAT: microcatheter assisted trabeculotomy; IT: inferior temporal; IN: inferior nasal; ST: superior temporal; SN: superior nasal.

**Figure 2 jcm-11-07279-f002:**
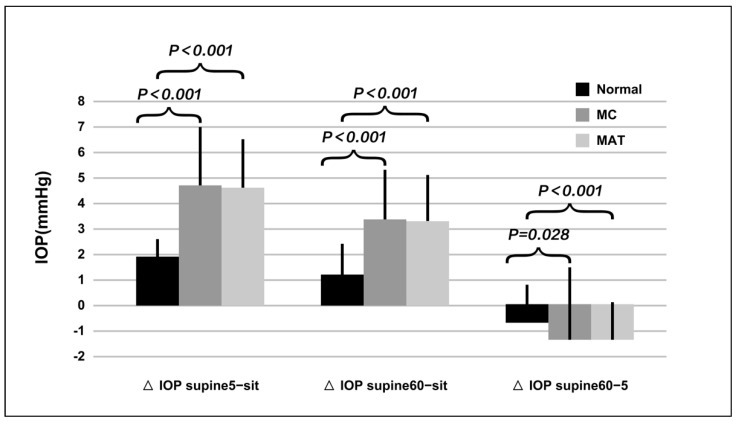
Comparison of intraocular pressure changes induced by postural alterations. Note: CP: canaloplasty, MAT: microcatheter assisted trabeculotomy. ΔIOP Supine 5-sit: The IOP difference between subjects lying for 5 min and in sitting position, ΔIOP Supine 60-sit: The IOP difference between subjects lying for 60 min and in sitting position, ΔIOP Supine 60-5: The IOP difference between subjects lying for 60 min and lying for 5 min.

**Figure 3 jcm-11-07279-f003:**
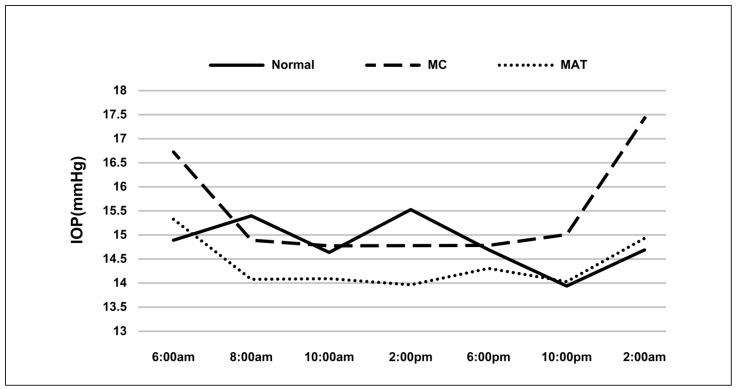
Diurnal IOP in different groups. Note: MC: modified canaloplasty; MAT: microcatheter assisted trabeculotomy, IOP: intraocular pressure.

**Table 1 jcm-11-07279-t001:** Demographic of subjects.

	Normal	MC	MAT	*p*
No. of eyes	20	25	30	
Age * (year),	41.2 ± 12.9	40.6 ± 12.0	41.0 ± 8.3	0.942
Gender (M/F)	12/8	14/11	18/12	0.406
OD/OS	14/6	15/10	14/16	0.199
Intervals (month)		18.8 ± 4.0	18.5 ± 3.1	0.527
Pre-IOP * (mmHg)		25.9 ± 5.0	24.4 ± 5.6	0.261
No. medication *		3.0 ± 0.5	3.1 ± 0.6	0.621
Post-IOP*visit1 (mmHg)	15.4 ± 1.6	14.8 ± 2.4	14.6 ± 2.3	0.452

* Quantitative data are shown as Mean ± Standard Deviation. MC: modified canaloplasty; MAT: microcatheter assisted trabeculotomy; Intervals: the time intervals between surgical procedures and the initiation of this study; IOP: intraocular pressure; Pre-IOP: IOP before surgery; Post-IOPvisit1: IOP measurement during the first clinic visit after the initiation of this study.

**Table 2 jcm-11-07279-t002:** Diurnal IOP (mmHg).

	Normal	Canaloplasty	MAT
IOP 2:00 am	15.4 ± 2.2	17.4 ± 3.1	15.0 ± 2.9
IOP 6:00 am	14.9 ± 2.0	16.7 ± 2.3	15.3 ± 3.0
IOP 8:00 am	14.7 ± 2.0	14.9 ± 2.1	15.0 ± 2.2
IOP 10:00 am	14.7 ± 2.2	14.5 ± 2.3	14.1 ± 2.1
IOP 2:00 pm	15.5 ± 1.7	14.8 ± 2.6	14.0 ± 2.3
IOP 6:00 pm	14.8 ± 2.0	14.8 ± 2.2	14.3 ± 2.3
IOP 10:00 pm	14.0 ± 1.9	15.0 ± 2.2	14.1 ± 2.3
IOP max	17.6 ± 1.9	18.6 ± 2.4	16.4 ± 2.4
IOP min	13.4 ± 1.7	13.3 ± 1.9	13.0 ± 2.1
ΔIOP	4.2 ± 1.2	5.2 ± 1.6	3.4 ± 1.2

MAT: microcatheter assisted trabeculotomy, IOP: intraocular pressure, Δ: change.

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
