# Peer review of "Intraocular Pressure Fluctuation in Primary Open-Angle Glaucoma with Canaloplasty and Microcatheter Assisted Trabeculotomy"

_jcm, 2022, doi:10.3390/jcm11247279_

Round 1

Reviewer 1 Report

In the methods  the visualization of the aqueous veins needs to be better explained with images.

In lines 139-140 is not clear wich is the difference of aqueous veins between normal and PAOG subject.

Table 2 is not present.

In the figure 3 the dotted lines (MC and Normal)  need to be better distinguished

Author Response

Respective  reviewer:

Thank you for the careful reading and thinking of our manuscript.  Following is the point-by-point responses to your comments.

1.In the methods  the visualization of the aqueous veins needs to be better explained with images.

Response: I have submitted the video of aqueous vein as supplement 2.

2.In lines 139-140 is not clear wich is the difference of aqueous veins between normal and PAOG subject.

Response: In normals, the pulsatile stratification of the aqueous and blood was observed in aqueous vein. In POAG the pulsatile oscillations becomes little or the aqueous vein is filled with blood only. 

3. Table 2 is not present.

Response: Thank you for pointing out the mistake. I have submitted the table 2.

4. In the figure 3 the dotted lines (MC and Normal)  need to be better distinguished.

Response: I modified the figure 3 as your suggestion.

Reviewer 2 Report

I have carefully reviewed the paper entitled “Intraocular Pressure Fluctuation in Primary Open-Angle Glaucoma with canaloplasty and microcatheter assisted trabeculotomy”. I have some concerns with regard the methods of this work.

o Provide surgical video for each of the used techniques.

o Could you also provide some of pictures showing and identifying aqueous veins?

o Did you put sutures on corneal wounds during surgery? You performed gonioscopy at a relatively early period after MAT and MC.

o Provide gonioscopic appearance of the blood reflux.

o Methods section is still confusing. When did you examine these patients? Two weeks after surgery? So, this study seems prospective in nature. Is that your routine follow-up style after intervention?

o Figure 2 > correct “supoine”

Author Response

Respective reviewer:

Thank you for the careful reading and thinking of our manuscript.  Following is the point-by-point responses to your comments.

1. Provide surgical video for each of the used techniques.

Responses: I added the videos of aqueous vein, videos of canaloplasty and MAT and pictures of blood reflux into SC with gonioscope as supplements.

2.Could you also provide some of pictures showing and identifying aqueous veins? Responses: I added the videos of aqueous vein as supplements 2. 3.Did you put sutures on corneal wounds during surgery? You performed gonioscopy at a relatively early period after MAT and MC. Responses: We did not putt sutures on corneal wounds. We performed the gonioscopy 1 year after the surgical procedures. 4.  Provide gonioscopic appearance of the blood reflux. Responses: I have added pictures of blood reflux into SC with gonioscope. 5. Methods section is still confusing. When did you examine these patients? Two weeks after surgery? So, this study seems prospective in nature. Is that your routine follow-up style after intervention? Responses: Thank you for the valuable suggestion. We recruited subjects with successful canaloplasty and MC more than 1 year ago. Thus it could be treated as a cross-sectional study. 6. Figure 2 > correct “supoine” Response: I corrected the figure 2 as your suggestion.

Reviewer 3 Report

Your study is interesting.

My queries are as below

1)     69: No “sighs“, literally is no “signs”

2)    According to your study method makes me wonder whether this should be “retrospective cohort” design or not as described, as to

-       The procedures of postural IOP check or aqueous vein, etc. are not ophthalmology routine, if they were done in the past, did you expect that for being a study?

-       If just a later recruitment, the patients were called up for a study, therefore, it’s likely to be a cross sectional analytical study

In terms of both circumstances, the informed consent from the patients should be mentioned.

As a result, please consider the detail and designate the study to get the most relevancy.

3)    Your subjects had mean age at around 40 at all groups, so when to do canaloplasty and MAT, were cases selected to get done? Please also describe the reasons if selected.

4)    Despite mean age are young patients, were cases with canaloplasty / MAT done as the sole procedure or there are cases combined with cataract surgery?

Thank you

Author Response

Respective reviewer:

Thank you for the careful reading and thinking of our manuscript.  Following is the point-by-point responses to your comments.

1. 69: No “sighs“, literally is no “signs”

Responses: I have corrected "sighs" to signs. 2. According to your study method makes me wonder whether this should be “retrospective cohort” design or not as described. Responses: I have clarified the methods part. As your suggestion, we just recruited subjects with successful surgical procedures more than 1 year ago and performed the ocular examination. So it should be the cross-sectional study.

3. Your subjects had mean age at around 40 at all groups, so when to do canaloplasty and MAT, were cases selected to get done? Please also describe the reasons if selected.

Responses: In this study we just collected the subjects receiving canaloplasty and MAT without any other surgical procedures, such as cataract surgery. And also MAT was reported better IOP control in young and middle-aged glaucoma than elder one. We also try to achieve age-matched between groups, because the IOP variation might be different in elder people. So the mean age of our subjects were relative young, around 40.

4. Despite mean age are young patients, were cases with canaloplasty / MAT done as the sole procedure or there are cases combined with cataract surgery? Responses: In this study, we just recruited cases with sole procedure, canaloplasty or MAT.  

Round 2

Reviewer 1 Report

I think it is necessary an image in addition to the video to better explain the aqueous veins. Table 2 is still missing in the latest version.

Author Response

Respective reviewer,

Thank you for reviewing our paper and providing valuable suggestion and comments.

I have added the table 2 in the revised version and added the figure of aqueous vein.

Best,

Chen 

Reviewer 2 Report

Especially thank for this brief and introductive videos. 

Author Response

Respective reviewer,

Thank you for reviewing our paper and providing valuable suggestion and comments. 

Best,

Chen

Reviewer 3 Report

none for now

Author Response

Respective reviewer,

Thank you for reviewing our paper and providing valuable suggestion and comments. I have looked through the manuscript and revised the spell mistakes.

Best,

Chen